# Multi-Omics Reveals Mechanisms of Partial Modulation of COVID-19 Dysregulation by Glucocorticoid Treatment

**DOI:** 10.3390/ijms232012079

**Published:** 2022-10-11

**Authors:** Matt Spick, Amy Campbell, Ivona Baricevic-Jones, Johanna von Gerichten, Holly-May Lewis, Cecile F. Frampas, Katie Longman, Alexander Stewart, Deborah Dunn-Walters, Debra J. Skene, Nophar Geifman, Anthony D. Whetton, Melanie J. Bailey

**Affiliations:** 1Faculty of Engineering and Physical Sciences, University of Surrey, Guildford GU2 7XH, UK; 2Stoller Biomarker Discovery Centre, School of Medical Sciences, Faculty of Biology, Medicine and Health, University of Manchester, Manchester M13 9NQ, UK; 3Faculty of Health and Medical Sciences, University of Surrey, Guildford GU2 7XH, UK; 4School of Health Sciences, University of Surrey, Guildford GU2 7XH, UK; 5School of Veterinary Medicine, School of Biosciences and Medicine, University of Surrey, Guildford GU2 7XH, UK

**Keywords:** glucocorticoid, dexamethasone, COVID-19, proteomics, metabolomics, mass spectrometry, multi-omics

## Abstract

Treatments for COVID-19 infections have improved dramatically since the beginning of the pandemic, and glucocorticoids have been a key tool in improving mortality rates. The UK’s National Institute for Health and Care Excellence guidance is for treatment to be targeted only at those requiring oxygen supplementation, however, and the interactions between glucocorticoids and COVID-19 are not completely understood. In this work, a multi-omic analysis of 98 inpatient-recruited participants was performed by quantitative metabolomics (using targeted liquid chromatography-mass spectrometry) and data-independent acquisition proteomics. Both ‘omics datasets were analysed for statistically significant features and pathways differentiating participants whose treatment regimens did or did not include glucocorticoids. Metabolomic differences in glucocorticoid-treated patients included the modulation of cortisol and bile acid concentrations in serum, but no alleviation of serum dyslipidemia or increased amino acid concentrations (including tyrosine and arginine) in the glucocorticoid-treated cohort relative to the untreated cohort. Proteomic pathway analysis indicated neutrophil and platelet degranulation as influenced by glucocorticoid treatment. These results are in keeping with the key role of platelet-associated pathways and neutrophils in COVID-19 pathogenesis and provide opportunity for further understanding of glucocorticoid action. The findings also, however, highlight that glucocorticoids are not fully effective across the wide range of ‘omics dysregulation caused by COVID-19 infections.

## 1. Introduction

Whilst great strides have been made in testing for and vaccinating against COVID-19, due to the rapid spread of new variants, the disease is progressing towards a globally endemic status. More positively, treatment of the illness has also improved, with mortality risk declining since the initial wave of COVID-19 infections in early 2020 [1]. A key treatment for COVID-19 infections is glucocorticoid medication, for example, dexamethasone. Synthetic glucocorticoids are a class of immunosuppressive drugs and additionally have powerful anti-inflammatory effects, an important benefit given that severe cases of COVID-19 are characterised by a hyperinflammatory state. The clinical action of glucocorticoids such as dexamethasone is well described for COVID-19 treatment, with the drug’s anti-inflammatory properties limiting lung injury, reducing both the likelihood of respiratory complications and also death [2]. It should be noted, however, that the UK’s National Institute for Health and Care Excellence (NICE) guidelines on dexamethasone recommend treatment only for those patients requiring supplemental oxygen, or with a level of hypoxia that needs supplemental oxygen but who are unable to have or tolerate such support [3].

Glucocorticoids act through trans-repression (the decreased expression of pro-inflammatory genes), trans-activation (the increased expression of anti-inflammatory genes) and also in part through non-genomic mechanisms, and have a long history in the treatment of chronic upper airways disease [4]. As with many other medications, glucocorticoids have side effects, including acne, indigestion, adrenal suppression and/or hepatotoxicity [5], as well as broader metabolomic, proteomic and clinical alterations such as eosinopenia [6]. These alterations have been reported to include changes in lipid metabolism in peripheral blood [7], with increased accumulation of triglycerides and sphingolipids [8]. Importantly, metabolic changes due to glucocorticoid treatment are complex and inter-related, rather than straightforward and unidirectional [9]. COVID-19 itself also influences the proteome and metabolome of those infected. The COVID-19 infection has been associated with alterations in amino acid serum levels, and also with lipids involved in glycerol metabolism, particularly triglycerides [10,11,12]. Bile acids have been shown to be dysregulated in COVID-19-positive patients and concentrations have been reported as decreased compared to negative patients [13]. There are a number of proteomic studies of COVID-19 patients showing some agreement between proteins identified as modulated due to this disease [14]. Inflammatory markers are elevated, particularly cytokines, often released by hematopoietic cells and platelets [15]. Both illness and treatment have independent and extensive influences on their host metabolomic and proteomic pathways, and the interactions between the two will in turn be complex and patient-specific.

Metabolomic and proteomic analyses of COVID-19-positive individuals have so far mainly concentrated on identifying biomarkers for prognosis and diagnosis [16,17,18], rather than the impact of different treatment regimens. Of the small number of studies to analyse dexamethasone treatment by ‘omics in COVID-19 patients (as opposed to clinical presentation), neutrophil degranulation has been identified as a modulated pathway [19]. Neutrophils are short-lived myeloid cells that act as a first line of defence against infection; this involves a degranulation process that is inclusive of cytokine release as well as proteolytic enzymes [20]. Neutrophils have also been reported to produce large quantities of neutrophil extracellular traps (that promote coagulation and inflammation), for example in patients suffering from COVID-19 acute respiratory distress syndrome (ARDS), a form of immune mechanism that is not modulated by dexamethasone treatment [21]. One study reviewed the possible impact of dexamethasone on COVID-19 patients by analysing transcriptomic data from studies on dexamethasone treatment in other conditions. As an indirect study, however, it was not able to offer insight into molecular pathological observations in COVID-19 patients [22]. To our knowledge, no study to date has examined the impact of glucocorticoids in COVID-19-positive participants using targeted and quantitative liquid chromatography-mass spectrometry (LC-MS), in order to better inform treatment choices.

The objective of this study was to quantitatively investigate the serum metabolome, lipidome and proteome of hospital inpatients with COVID-19 to identify the combined influence of glucocorticoid treatment and COVID-19 infection, benefiting from the additional insight offered by combining multiple ‘omics approaches [23,24]. Because glucocorticoids were not prescribed for SARS-CoV-2 infections during the initial wave of COVID-19 in the UK (March to June 2020) but were widely prescribed in the second wave (July 2020 onwards), there is a time component which was not controlled for in this retrospective observational study, including the presence in the UK of new variants [25,26,27]. For metabolomics, the Biocrates quantitative metabolomics platform was used, employing tandem mass spectrometry (MS/MS) [28]. For proteomics, analysis was performed using the SWATH-MS technique, implementing a Data-Independent Acquisition (DIA) approach for precision identification and accurate relative quantitation of proteins [29].

The results presented here are consistent with previous work that the glucocorticoid dexamethasone modulates the neutrophil response, and also modulates cortisol levels and bile acid dysregulation. The data also, however, illustrate that in many cases, glucocorticoid treatment is not sufficient to reverse the observed phenomena of the impact of COVID-19, for example in compensating for dyslipidemia or elevated amino acid levels in serum.

## 2. Results

### 2.1. Population Metadata Overview

The study population analysed in this work totalled 98 hospital inpatients recruited between May 2020 and March 2021. These included 37 participants with a positive COVID-19 RT-PCR test and treated with glucocorticoids up to 48 h prior to sampling, and 36 participants with a positive RT-PCR test but not treated with glucocorticoids. A control group (*n* = 25) with a negative COVID-19 RT-PCR test but with symptoms consistent with a suspected COVID-19 infection was also recruited, in a shorter timeframe between May 2020 and July 2020. Of the 37 participants treated with synthetic glucocorticoids, 30 were treated with dexamethasone, 6 with prednisolone, 1 with methylprednisolone and none with betamethasone or hydrocortisone. A summary of the population characteristics is shown in Table 1.

Age distributions for the glucocorticoid-treated COVID-19-positive cohort and the untreated cohort were similar, but the treated cohort included proportionately more males. Comorbidities are associated with severity, representing both a causative and confounding factor. Due to hospital recruitment, however, comorbidities including type 2 diabetes mellitus, hypertension, high cholesterol and ischaemic heart disease were present in both the treated and untreated groups; former smokers were somewhat more represented in the glucocorticoid-treated group. Levels of C-Reactive Protein (CRP) and eosinophils were reduced by glucocorticoid treatment (*p*-values of 0.03 and 0.02, respectively), but levels of lymphocytes were not. Within the treated cohort, participants were more likely to present with bilateral chest X-ray changes (*p*-value of 0.001), more likely to require continuous positive airways pressure (*p*-value of 0.008) and were also escalated to the hospital Medical Acute Dependency Unit more frequently (*p*-value of 0.001), indicating that those treated with glucocorticoids are more severely affected by COVID-19 than those not treated.

### 2.2. Feature Identification

We identified 472 serum metabolites and lipids that were reliably quantified in samples (out of a theoretical maximum of 630). We then identified those metabolites that were differentiated between those treated with glucocorticoids and those not treated at a *p*-value of 0.05 or less by the non-parametric Wilcoxon rank sum test. This resulted in a list of 53 metabolites and lipids significantly altered by glucocorticoid treatment.

For proteomics, SWATH-MS identified 754 proteins. This initial dataset was then filtered by the same method as for metabolites in order to identify proteins differentially expressed between glucocorticoids-treated and -untreated participants. A total of 68 proteins were found to be modulated.

The top 15 altered metabolites (Table 2) and proteins (Table 3) ranked by p-value are summarised below. Complete lists of differentiated proteins and metabolites are shown in Appendix A, respectively; the full lists of proteins and metabolites identified are shown in Appendix A.

### 2.3. Metabolomic Analysis: Key Metabolites Altered by Glucocorticoid Treatment

The initial data frame of metabolites measured in participants was subjected to pathway analysis. The most statistically significant pathway when controlled for false discovery was steroid hormone biosynthesis (Table 4). Several amino acid pathways also showed statistically significant differences, including lysine, phenylalanine, tyrosine and tryptophan.

Univariate analyses by boxplot for cortisol, bile acids and amino acids (selected for their relevance to the statistically significant pathways in Table 4 or their prior literature identification relevant to COVID-19 infections) are also shown in Figure 1. Fewer lipids were given categorical identifiers in the pathway analysis described above due to assay and database limitations. Statistically significant triglycerides are also shown in Figure 1.

### 2.4. Proteomic Analysis: Investigation of Pathway Changes Due to Glucocorticoid Treatment

Next, differentially expressed proteins were used for pathway analysis. The results of the pathway analysis as performed in ClueGO are summarised in Table 5. The most statistically significant set of pathways when controlled for false discovery were neutrophil degranulation and the innate immune system, followed by exocytosis of platelets, platelet activation, signalling and degranulation. A summary of the affected pathways is also presented in network form in Figure 2.

Univariate analyses by boxplot for significantly altered proteins are also shown in Figure 3, showing the treated and untreated groups, as well as the control group (COVID-19 negative, no glucocorticoid treatment) for comparison.

### 2.5. Relevance Network Analysis of Glucocorticoid Treatment

Finally, a relevance network analysis base was performed on the proteomic and metabolomic datasets, using a bipartite implementation applied to identify related features. Figure 4 shows these related features as nodes (proteins and metabolites) together with the links between them (similarity scores of greater than 0.65). Eight proteins and 20 metabolites were identified in this relevance network.

Lipids incorporating the fatty acid arachidonic acid (denoted 20.4 equalling 20:4n6) were over-represented in the relevance network, comprising 8 of the 20 metabolites identified. In the overall feature set, lipids incorporating arachidonic acid comprised 13 of the 483 metabolites measured by the targeted LC-MS/MS method used in this work.

## 3. Discussion

We have employed a quantitative multiomics approach combined with use of clinical data to investigate glucocorticoid action in patients with COVID-19 across metabolomic and proteomic features. Metabolomic pathway analysis shows that the largest difference between the glucocorticoid-treated and -untreated groups was in steroid hormone biosynthesis, as would be expected. Cortisol levels in the treated group were below those seen in either the untreated group, or the control group. The remaining pathways influenced were predominantly related to amino acids. Amino acid dysregulation was greater in glucocorticoid-treated participants than in -untreated participants. Consequently, these data show no evidence of glucocorticoids modulating COVID-19-driven amino acid dysregulation. Furthermore, glucocorticoid treatment did not alleviate increased serum concentrations of triglycerides. This result is consistent with dexamethasone previously being identified as potentially causing hypertriglyceridemia in non-COVID-19 settings [7], and suggests that in part, the effect of COVID-19 and glucocorticoid treatment on dyslipidemia may be additive. These data illustrate the complexity of drug/disease interactions.

For the proteomic dataset, the most statistically significant pathway change caused by glucocorticoid treatment was that for neutrophil degranulation. This pathway is related to platelet activation and degranulation in an inflammatory response. The importance of neutrophil- and platelet-associated pathways described here is consistent with the COVID-19 proteomics literature, especially Sinha et al. and Panda et al. [19,21], and reveals the molecular mechanisms for synthetic glucocorticoids alleviating COVID-19 symptoms. Of interest is the fact that within these protein pathway alterations, the impact of glucocorticoids was not consistent, with some proteins moderated towards the levels seen in the control group whilst others were not, or indeed showed amplified changes.

The relevance network presented in this work (Figure 4) also shows relationships that have previously been related to COVID-19 infections, albeit care is required in interpreting relevance networks as they cannot show causation. For example, P04196 has a role in immune complex and pathogen clearance, cell chemotaxis, cell adhesion, angiogenesis, coagulation and fibrinolysis. Its modulated expression and relationship to cortisol fits with a dampening of the innate and acquired immune system. P35542 or Serum Amyloid A4 Protein is critical in the acute phase response to infection and/or injury and, as such, it is not a surprise to see it is modulated by dexamethasone. Q96IY4 inhibits thrombolytic response via specific proteolytic catalysis. Again, this is in keeping with glucocorticoids moderating the innate and acquired response to infection. Furthermore neutrophil activation can be caused by elevated free fatty acids and triglycerides [30], as well as GM-CSF (a cytokine elevated in COVID-19). GM-CSF expression is abrogated by dexamethasone [31,32]. Additionally, triglycerides incorporating arachidonic acid were related to the proteins P25311, Q96IY4, P22532 and P10909. Arachidonic acid has been identified as a potential marker of severe COVID-19 infection, acting as a substrate for the lipoxygenase and cyclooxygenase pathways, which contribute to increased levels of eicosanoids [33,34]. Whilst the disproportionate presence of arachidonic acid in differentiating lipids cannot be causatively linked to leukotriene or prostaglandin expression, arachidonic acid-derived oxylipins are generally viewed as pro-inflammatory [35], and have been associated with the immune response to COVID-19 [36]. Furthermore, patients with hypertriglyceridemia also demonstrate increased platelet activation [37]. Thus, there may well be a relationship between increased triglycerides and platelet activation in COVID-19 which is not being alleviated by treatment with glucocorticoids.

It should further be noted that the World Health Organization recommends the use of dexamethasone only in severe/critical patients, and not routinely, as do the UK’s NICE guidelines. More recently, clinical data have shown that treatment with dexamethasone in patients that do not require intensive respiratory support is not associated with any improvement in outcomes, and that there is evidence of potential harm in such cases [38]. We now provide ‘omics-driven evidence of the potential pleiotropic actions of synthetic glucocorticoids.

This study does include clear limitations. As a retrospective observational study, results will naturally be less robust than those that would be obtained from a prospective randomized and controlled trial. Furthermore, dexamethasone became standard of care in the UK in June 2020, albeit the implementation was not uniform [39], which meant that the cohort not treated by glucocorticoids had a time difference to the treated group. As discussed previously, whilst all participants were recruited in an in-patient hospital setting and there were no differences in admittance to ICU, there was a difference in treatment with CPAP and MADU admittance, strongly suggestive of the glucocorticoid-treated cohort having more severe symptoms. Other standards of care (ventilation, anti-clotting drugs) were also changing over the recruitment period. In addition, participants were partially recruited at a time when new variants were beginning to emerge, specifically the Alpha variant, which was present in the UK from January to April 2021, and the Delta variant, which was present in the UK from April to December 2021 [40]. This may have led to differences in underlying severity or clinical presentation. Furthermore, samples in this work were not sequenced so cannot be directly mapped to their variants. It should be noted that multi-omics can include the full range of genomics, transcriptomics, proteomics and metabolomics, but in this work, proteomics and metabolomics were employed. Finally, these analyses were conducted in a pandemic setting, where the circulation of competing respiratory viruses was limited. The specificity of the findings here to COVID-19 (as opposed to other respiratory diseases) has not been tested.

In conclusion, these results provide the first multi-omics investigation into the action of synthetic glucocorticoids in COVID-19 treatment, and are concordant with previous findings that glucocorticoids modulate the neutrophil response [19]. The findings are also consistent with clinical observations of reduced hyperinflammatory reactions [41], potentially limiting immune system-related harm to the respiratory system. The data do, however, illustrate that in many cases, glucocorticoid treatment will not fully alleviate the impact of COVID-19. In some instances, it is possible that the previously described use of dexamethasone causing hypertriglyceridemia may also play a role in this observation, albeit it is not possible in this work to separate the impact of glucocorticoids from the higher severity of COVID-19 in the treated cohort. Whilst glucocorticoids have been demonstrated to be effective in the treatment of severe cases of COVID-19, as with any class of drugs, caution is needed in deeming any treatment regimen as suitable for universal use.

## 4. Materials and Methods

### 4.1. Participant Recruitment and Ethics

All participants for this study were recruited as part of an observational cohort study. Ethical approval (IRAS project ID 155921) was obtained via the NHS Health Research Authority (REC reference: 14/LO/1221). The participants were recruited consecutively at Frimley Park NHS Trust, UK, between May 2020 and March 2021. Participants were identified by clinical staff to ensure that they had the capacity to consent to the study and were asked to sign an Informed Consent Form based on the International Severe Acute Respiratory and emerging Infection Consortium/World Health Organisation (ISARIC/WHO) Clinical Characterisation Protocol for Severe Emerging Infections. Those patients that did not have this capacity were not sampled. Signatures were witnessed by University of Surrey researchers. At the time of recruitment, participants were categorised by the hospital as either “query COVID” (meaning there was clinical suspicion of COVID-19 infection, but a negative positive RT-PCR SARS-CoV-2 test result had been recorded during their admission) or “COVID positive” (meaning that a positive test result had been recorded). All participants were provided with a Patient Information Sheet explaining the goals of the study. All methods part of this study were performed in accordance with the relevant guidelines and regulations.

### 4.2. Sample Collection and Extraction

Collection of the samples was performed by researchers from the University of Surrey at Frimley Park NHS Foundation Trust hospitals; collection took place on admission or in some cases shortly afterwards. Alongside the collection of blood samples, metadata for all participants were also collected, covering the inter alia medication regime (specifically including dexamethasone treatment or other glucocorticoids), sex, age, comorbidities (based on whether the participant was receiving treatment), the results and dates of COVID-19 PCR (polymerase chain reaction) tests, bilateral chest X-Ray changes, smoking status and whether the participant presented with clinical symptoms of COVID-19. Values for lymphocytes, CRP and eosinophils were also taken—here, values within five days of biofluid sampling were recorded.

Serum collection and extraction followed the protocols set out by the COVID-19 Coalition for metabolomics [42] and proteomics [43]. In brief, venous blood was collected in 3 mL serum tubes, transported to the University of Surrey by courier whilst stored on ice, and centrifuged on arrival at 1600× *g* for 10 min at 4 °C. All samples with a sampling time interval greater than four hours were rejected. Serum was then decanted into 100 µL aliquots and stored at −80 °C until processing. Prior to analysis, the serum was sterilised using 200 µL of ethanol into 100 µL of serum (2:1 *v*/*v* solvent/sample ratio).

### 4.3. Participant Selection

From the initial population of 115 recruited participants, a number of exclusions were made to remove participants where incomplete metadata were provided, where the gap between the initial positive COVID-19 test and sampling exceeded 14 days or where the participants were already being medicated with glucocorticoids prior to COVID-19 treatment. The remaining population of 98 was then segmented into a COVID-19-negative control group for comparison purposes (*n* = 25), a group of COVID-19-positive patients treated with a glucocorticoid in the 48 h prior to sampling (*n* = 37), and a group of COVID-19-positive patients not treated with a glucocorticoid in in the 48 h prior to sampling (*n* = 36).

### 4.4. Serum Instrumentation and Analysis: Metabolomics

Serum samples were analysed using the Biocrates MxP Quant 500 system using a Xevo TQ-S Triple Quadrupole Mass Spectrometer coupled to an Acquity UPLC system (Waters Corporation, Milford, MA, USA). The MxP Quant 500 system provides targeted quantification of metabolites including amino acids and derivatives, bile acids, biogenic amines, acylcarnitines, carbohydrates and other small molecule metabolites, plus a wide array of lipids. Analysis took place via a single assay, and two analytical procedures. The first of these procedures operated by liquid chromatography (operated in both positive and negative ion mode) and the second by flow injection analysis (positive ion mode), both coupled to tandem mass spectrometry with isotopically labelled internal standards for quantification. The sample order was randomised across 96-well plates, and 3 levels of quality controls (QC) were run on each plate. Blank PBS (phosphate-buffered saline) samples (three technical replicates) were used for the calculation of the limits of detection (LOD). Biogenic amines and amino acids were quantified for each plate using a seven-point calibration curve, with other analytes semi-quantitated with a single point standard (i.e., assuming concentration linearity in the range measured). The levels of metabolites present in each QC were compared to the expected values and the CV% calculated. Data were normalised between the three batches using the results of quality control level 2 (QC2) repeats across the plate (*n* = 5) and between plates (*n* = 3) using Biocrates METIDQ software (QC2 correction). Metabolites where >25% concentrations were at or below the limit of detection (≪LOD), above the limit of quantification (>LOQ), or where the blank was out of range were excluded (total n excluded in serum = 150). The remaining 474 quantified metabolites comprised of 8 acylcarnitines, 20 amino acids, 26 biogenic amines, 11 bile acids, 53 ceramides, 15 cholesteryl esters, 1 cresol, 9 diglycerides, 4 carboxylic and fatty acids, 85 phosphatidyl cholines, 14 sphingolipids, 222 triglycerides, 2 hormones, 2 indoles, 1 nucleobase and 1 vitamin.

### 4.5. Plasma Instrumentation and Analysis: Proteomics

Mass spectrometry of clinical specimens using the SWATH-MS technique implements a Data-Independent Acquisition (DIA) approach for precision identification and accurate quantification of proteins. Samples were quality-checked, assigned a unique in-ternal study ID and groups were randomized. Immunodepletion was performed by using Top14 Abundant Protein Depletion Resin (ThermoFisher Scientific, Macclesfield, UK) in a 96-well format. Samples were reduced, alkylated and digested with trypsin (Promega, Macclesfield, UK) using S-trap columns (Protifi, New York, NY, USA) prior to lyophylisation. Digitized proteomic maps were generated by using 100 variable window SWATH-MS (68 min) with a micro-flow LC-MS system. SWATH-MS analysis was performed on a 6600 TripleTOF mass spectrometer with DuoSpray Ion Source (AB Sciex Limited, Alderley Park, Macclesfield, UK) coupled to an Eksigent 425 LC (AB Sciex Limited) with specific mass spectrometric conditions (including isolation window size and overlap and total cycle time), as previously described. All study samples were run in duplicate and CV percentages were calculated across the two injections. Only samples with median CV’s across all proteins quantified with less than 20% were taken forward for analysis, with the best injection for each sample included in the final protein quantification. To ensure instrument performance remained consistent throughout the study and to control for batch-related effects, both commercially available plasma (Human K2EDTA gender pooled plasma, Sera Laboratories International Ltd. trading as BioIVT, Burgess Hill, UK) and a total pool of all study samples (TOTs) were processed, digested and run alongside each batch of samples.

### 4.6. Feature Identification

Serum metabolites were identified and quantified using isotopically labelled internal standards, retention times and multiple reaction monitoring. In this study, identifications were made in accordance with the Metabolomics Standards Initiative for metabolite identification [44], using both accurate *m*/*z* values referenced to library values as well as orthogonal information in the form of MS/MS fragmentation spectra and retention time matching against isotopically labelled internal standards. The mass spectrometry conditions were used as optimised and provided by Biocrates.

For the proteomic workflow, extraction and processing followed the method described in Salie et al. [45]. Briefly, SWATH-MS data files were searched using openSWATH (version 2.0.0) against a published plasma spectral library [46]. Spectral library matches were scored and filtered for quality using pyProphet (version 0.18.1) then aligned across runs using the TRIC alignment algorithm from MSproteomicstools. The transition-level quantification contained within the aligned data was then normalised and summarised to protein-level quantification in R (version 3.4.1) using the Bioconductor (version 3.5) packages SWATH2stats and MSstats, with the equalized medians and top 3 features per peptide parameters. Proteins quantified in at least 30% of samples were retained in the following biomarker analysis.

The two approaches described generated two data frames, each comprising a matrix of n participants by p features, with matched n of 98 participants. These data frames are provided in Appendix A.

### 4.7. Statistical Analysis

Both the metabolomic and proteomic data frames were log2 transformed to stabilize variance and reduce heteroscedasticity. In order to identify proteins and metabolites differentially expressed between the two conditions (COVID-19-positive participants treated with glucocorticoids, and not treated), the difference between the two populations for each feature was measured by *p*-value. Specifically, a *p*-value threshold of 0.05 was used for significance by Wilcoxon rank-sum test.

Pathway analysis for the proteomic differences between the two classes of participants was conducted using the ClueGo application (version 2.5.9) within Cytoscape (version 3.9.1) [47,48], matching Uniprot identifiers for differentiating proteins to Reactome pathway databases. Pathway analysis for the metabolomic differences between the two classes of participants was conducted using MetaboAnalyst (version 5.0) [49], using the Pathway Analysis module, matching HMDB identifiers to the MetaboAnalyst database. Participant metadata characteristics for the two main cohorts (COVID-19 positive split between glucocorticoid-treated and -untreated patients) were assessed by two-tailed Student t-tests for continuous variables, and by two-tailed z-score tests for population proportions.

The relevance network was constructed using the R package MixOmics [50]. This approach uses the two data frames (metabolomic and proteomic) to construct a multiblock partial least squares discriminant analysis (PLS-DA) model. This model was then used to generate a relevance network for the variables selected by the multiblock PLS-DA, to highlight differential relationships between the treated cohort and the untreated (by glucocorticoid) cohort. Rather than a matched-pairs correlation network, a bipartite model was constructed, showing protein-to-metabolite relationships and excluding protein-to-protein/metabolite-to-metabolite relationships, using a similarity matrix to capture the relationships between the features and components of the predictive multiblock PLS-DA model [51]. The relevance network output file was then processed in Cytoscape to produce the visual summary.

## Figures and Tables

**Figure 1 ijms-23-12079-f001:**
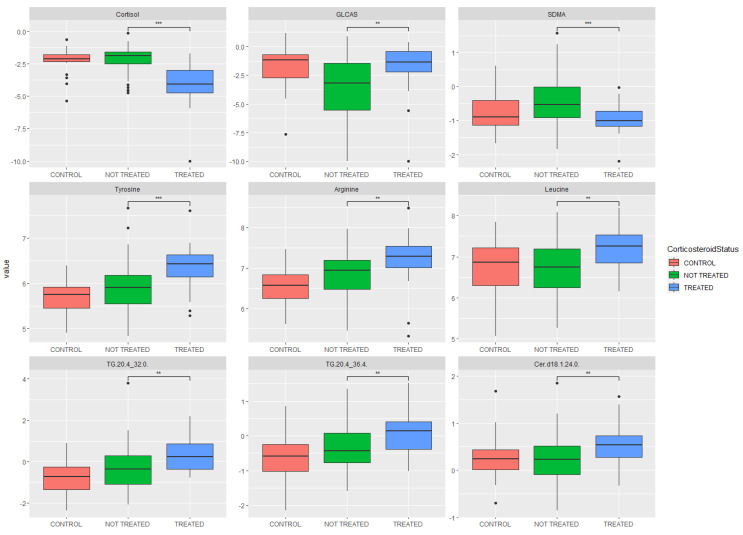
Boxplots of significantly altered metabolites. Plots show a control group for comparison (COVID-19 negative, not treated with glucocorticoids), COVID-19 positive (not treated with glucocorticoids) and COVID-19 positive (treated with glucocorticoids). GLCAS = glycolithocholic acid sulfate, TG = triglyceride, SDMA = symmetric dimethylarginine, Cer = ceramide. Statistical significance is shown between treated and untreated cohorts.** indicates *p* ≤ 0.01, *** indicates *p* ≤ 0.001.

**Figure 2 ijms-23-12079-f002:**
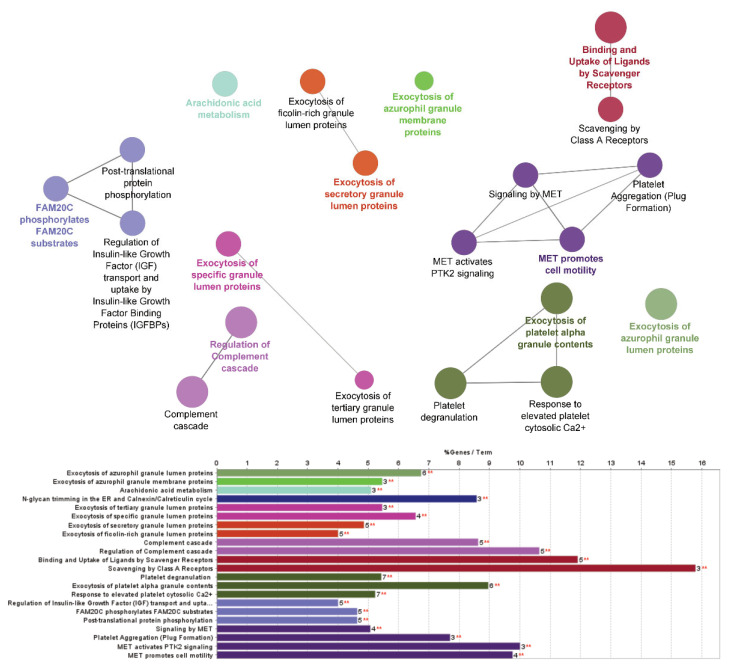
Network analysis of differentiated pathways. Figure includes both a network representation of pathways differentiated between COVID-19-positive participants treated with glucocorticoids and those not treated with glucocorticoids, together with the number of pathways altered (bottom chart) and false-discovery corrected statistical significance, ** indicates *p* ≤ 0.01 generated in CytoScape with ClueGO. Links between nodes are presented for readability and are not proportional to significance or impact.

**Figure 3 ijms-23-12079-f003:**
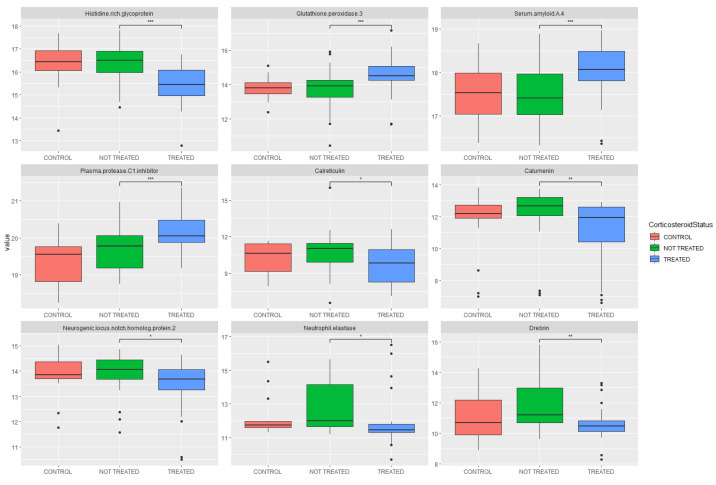
Boxplots of significantly altered metabolites. Plots show a control group for comparison (COVID-19 negative, not treated with glucocorticoids), COVID-19 positive (not treated with glucocorticoids) and COVID-19 positive (treated with glucocorticoids). Statistical significance is shown between treated and untreated cohorts. * indicates *p* ≤ 0.05, ** indicates *p* ≤ 0.01, *** indicates *p* ≤ 0.001.

**Figure 4 ijms-23-12079-f004:**
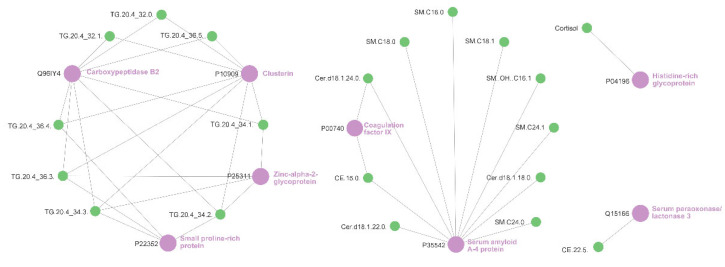
Relevance network analysis of features. The network shows those features identified as differentially expressed between COVID-19-positive participants treated with glucocorticoids and those not treated with glucocorticoids. Metabolites are shown as green nodes and proteins are shown as purple nodes. Generated in CytoScape from a network produced in the MixOmics R package. Links between nodes are presented for readability and are not proportional to significance or impact. TG = triglyceride, Cer = ceramide, SM = sphingomyelin, CE = cholesterol ester.

**Table 1 ijms-23-12079-t001:** Characteristics of study population. P-values are shown for the positive for COVID-19: treated with glucocorticoids group compared with the positive for COVID-19: not treated with glucocorticoids group.

Parameters	Control Group: Negative for COVID-19	Not Treated with Glucocorticoids: Positive for COVID-19	Treated with Glucocorticoids: Positive for COVID-19	*p*-Value Glucocorticoids versus Not Treated
*n*	25	36	37	
Age (mean, standard deviation; years)	65.8 ± 21.4	62.8 ± 21.2	60.6 ± 15.2	0.61
Male / Female (*n*)	11/14	20/16	26/11	0.19
Treated with Glucocorticoids (*n*)	0	0	37	0.00
Treated with Anticoagulants (*n*)	3	5	6	0.78
Treated for Hypertension (*n*)	8	15	15	0.92
Treated for High Cholesterol (*n*)	6	2	6	0.14
Treated for Type 2 Diabetes Mellitus (*n*)	8	16	8	0.04
Treated for Ischaemic Heart Disease (*n*)	6	5	6	0.78
Ex-Smoker (*n*)	9	9	16	0.10
Current Smoker (*n*)	0	2	0	0.15
Medical Acute Dependency admission (*n*)	4	10	24	0.001
Intensive Care Unit admission (*n*)	1	0	2	0.16
Did Not Survive Admission (*n*)	1	3	1	0.29
Duration of pre-admission symptoms (mean, standard deviation; days)	6.9 ± 10.5	4.1 ± 4.1	7.0 ± 4.3	0.004
Time between positive RT-PCR test and sampling (mean, standard deviation; days)	na	4.3 ± 5.7	3.2 ± 3.8	0.35
Lymphocytes (mean, standard deviation; cells/μL)	0.8 ± 0.4	0.8 ± 0.6	0.7 ± 0.6	0.36
C-Reactive Protein (mean, standard deviation; mg/L)	129.8 ± 94.2	127.5 ± 97.8	96.3 ± 72.2	0.13
Eosinophils (mean, standard deviation; 100/μL)	0.3 ± 0.3	0.1 ± 0.1	0.0 ± 0.0	<0.001
Bilateral Chest X-Ray changes (*n*)	2	12	27	0.001
Continuous Positive Airway Pressure (*n*)	3	4	14	0.008

**Table 2 ijms-23-12079-t002:** Top 15 metabolites ranked by p-value between treated and untreated cohorts measured by the Wilcoxon rank sum non-parametric test. Cer = ceramide, CE = cholesterol ester, SM = sphingomyelin.

Metabolite	*p*-Value	Fold Change	Increased/Decreased in Glucocorticoid Group
Cortisol	9.27 × 10^−9^	0.26	Decreased
α-Aminoadipic acid	4.79 × 10^−4^	1.68	Increased
Tyrosine	7.95 × 10^−4^	1.31	Increased
Propionylcarnitine	7.95 × 10^−4^	1.66	Increased
Cer (d18:1/22:0)	8.61 × 10^−4^	1.28	Increased
Phenylalanine	9.69 × 10^−4^	1.32	Increased
CE (15:0)	1.22 × 10^−3^	1.27	Increased
Aconitic acid	1.27 × 10^−3^	0.70	Decreased
Cer (d18:1/24:0)	1.37 × 10^−3^	1.28	Increased
SM C18:0	1.69 × 10^−3^	1.30	Increased
Carnitine	2.00 × 10^−3^	1.34	Increased
Arginine	6.86 × 10^−3^	1.28	Increased
Symmetric dimethylarginine	6.98 × 10^−3^	0.72	Decreased
Methionine	7.46 × 10^−3^	1.23	Increased
Cer (d18:2/24:0)	7.58 × 10^−3^	1.31	Increased

**Table 3 ijms-23-12079-t003:** Top 15 proteins ranked by *p*-value different between treated and untreated cohorts measured by the Wilcoxon rank sum non-parametric test.

Protein	Uniprot ID	*p*-Value	Fold Change	Increased/Decreased in Glucocorticoid Group
Histidine-rich glycoprotein	P04196	3.01 × 10^−5^	0.55	Decreased
Glutathione peroxidase 3	P22352	3.31 × 10^−4^	1.84	Increased
Serum amyloid A-4 protein	P35542	6.76 × 10^−4^	1.44	Increased
Plasma protease C1 inhibitor	P05155	8.02 × 10^−4^	1.35	Increased
Calreticulin	P27797	1.39 × 10^−3^	0.46	Decreased
Calumenin	O43852	2.20 × 10^−3^	0.50	Decreased
Laminin subunit beta-1	P07942	2.47 × 10^−3^	0.85	Decreased
Neurogenic locus notch homolog protein 2	Q04721	2.71 × 10^−3^	0.74	Decreased
Neutrophil elastase	P08246	3.19 × 10^−3^	0.54	Decreased
Serpin B3	P29508	3.22 × 10^−3^	0.61	Decreased
Drebrin	Q16643	3.76 × 10^−3^	0.46	Decreased
Coagulation factor IX	P00740	4.22 × 10^−3^	1.30	Increased
Cytoplasmic aconitate hydratase	P21399	5.35 × 10^−3^	0.78	Decreased
Transgelin	Q01995	5.50 × 10^−3^	0.75	Decreased
Carboxypeptidase B2	Q96IY4	5.82 × 10^−3^	1.35	Increased

**Table 4 ijms-23-12079-t004:** Metabolomics: pathways identified as differentially expressed between COVID-19-positive participants treated with glucocorticoids and those not treated with glucocorticoids.

Pathway Name	# Entities Identified	*p*-Value	*p*-Value FDR Corrected
Steroid hormone biosynthesis	2	5.45 × 10^−9^	2.29 × 10^−7^
Phenylalanine, tyrosine and tryptophan biosynthesis	2	6.89 × 10^−4^	9.65 × 10^−3^
Phenylalanine metabolism	2	6.89 × 10^−4^	9.65 × 10^−3^
Tyrosine metabolism	1	3.80 × 10^−3^	3.19 × 10^−2^
Ubiquinone and other terpenoid-quinone biosynthesis	1	3.80 × 10^−3^	3.19 × 10^−2^
Aminoacyl-tRNA biosynthesis	20	6.62 × 10^−3^	4.63 × 10^−2^
Arginine biosynthesis	3	1.20 × 10^−2^	5.70 × 10^−2^
Valine, leucine and isoleucine biosynthesis	4	1.20 × 10^−2^	5.70 × 10^−2^
Sphingolipid metabolism	1	1.22 × 10^−2^	5.70 × 10^−2^
Valine, leucine and isoleucine degradation	4	1.43 × 10^−2^	6.00 × 10^−2^

**Table 5 ijms-23-12079-t005:** Proteomics: pathways identified as differentially expressed between COVID-19-positive participants treated with glucocorticoids and those not treated with glucocorticoids.

Pathway Name	# Entities Identified	*p*-Value	*p*-Value FDR Corrected
Neutrophil degranulation	17	1.34 × 10^−8^	1.75 × 10^−7^
Innate immune system	23	1.34 × 10^−8^	1.75 × 10^−7^
Exocytosis of platelet alpha granule contents	6	2.75 × 10^−6^	3.30 × 10^−5^
Platelet activation, signalling and aggregation	10	2.75 × 10^−6^	3.30 × 10^−5^
Binding and uptake of ligands by scavenger receptors	5	4.32 × 10^−6^	4.75 × 10^−5^
Regulation of complement cascade	5	2.16 × 10^−5^	1.95 × 10^−4^
Platelet degranulation	7	2.75 × 10^−6^	3.30 × 10^−5^
Exocytosis of azurophil granule lumen proteins	6	1.29 × 10^−5^	1.29 × 10^−4^
Response to elevated platelet cytosolic Ca2+	7	2.75 × 10^−6^	3.30 × 10^−5^
Complement cascade	5	2.16 × 10^−5^	1.95 × 10^−4^

## Data Availability

Data frames for proteomics and metabolomics relating to this work are included within Appendix A.

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
