# Peer review of "Multi-Omics Reveals Mechanisms of Partial Modulation of COVID-19 Dysregulation by Glucocorticoid Treatment"

_ijms, 2022, doi:10.3390/ijms232012079_

Round 1

Reviewer 1 Report

in the discussion page 10 it should SAA type 4

Author Response

In the discussion page 10 it should SAA type 4

We have amended the manuscript text to read 'P35542 or Serum Amyloid A4 Protein' at the reviewer's suggestion. We thank the reviewer for this small but nonetheless helpful correction.

Reviewer 2 Report

The manuscript doesn't add new informations to international literature

Author Response

The manuscript doesn't add new information to international literature

We assure the reviewer that we have searched the literature to find 'omics studies of glucocorticoid action in COVID-19 positive participants, using real-world samples, and believe that the work is novel. If the reviewer has a specific paper in mind that already contains the information reported in our manuscript, we would be pleased to review it.

Reviewer 3 Report

This work by Matt Spick et al. provides the first multi-omics (metabolomic and proteomic) study of the action of synthetic glucocorticoids in COVID-19 treatment, based on clinic data from patients between 2020 and 2021 in the UK. Metabolomic pathway analysis reveals that the steroid hormone biosynthesis and amino acid are the two most affected parts. And proteomic analysis shows neutrophil degranulation is the most affected pathway. Nevertheless, the results show that glucocorticoid treatment does not fully alleviate the impact of COVID-19. Considering dexamethasone is used to treat severe patients, caution is needed in patient treatment using glucocorticoids. I think this study is necessary, and the knowledge should be available to the public. Thus, I support the publication of this work after they answer my following concerns. 

1. Glucocorticoid is the key molecule in this work. The molecular structure should be provided. 

2. Although this work focuses on studying and analyzing the effect of glucocorticoids on COVID-19 patients, the molecular mechanism of how the molecule affects can be mentioned.

3. As the authors already mentioned, the SARS-CoV-2 variant may affect the results. Their data were collected at the time when new variants (such as alpha and delta) showed in the UK. Thus, I suggest this issue and related references(1-3) should be mentioned in the introduction, not (only) at the end. 

4. For Figure 4, the items can be arranged in a better order and shape.

(1)  Harvey, W. T.; Carabelli, A. M.; Jackson, B.; Gupta, R. K.; Thomson, E. C.; Harrison, E. M.; Ludden, C.; Reeve, R.; Rambaut, A.; Consortium, C.-G. U. et al. Sars-cov-2 variants, spike mutations and immune escape. Nat. Rev. Microbiol. 2021, 19, 409-424.

(2)  Tian, F.; Tong, B.; Sun, L.; Shi, S.; Zheng, B.; Wang, Z.; Dong, X.; Zheng, P. N501y mutation of spike protein in sars-cov-2 strengthens its binding to receptor ace2. Elife 2021, 10, e69091.

(3)  Yurkovetskiy, L.; Wang, X.; Pascal, K. E.; Tomkins-Tinch, C.; Nyalile, T. P.; Wang, Y.; Baum, A.; Diehl, W. E.; Dauphin, A.; Carbone, C. et al. Structural and functional analysis of the d614g sars-cov-2 spike protein variant. Cell 2020, 183, 739-751.e8.

Author Response

We thank the reviewer for the overall positive response, and also for the helpful suggestions. We have incorporated all of these within the updated manuscript, and agree that they represent an improvement to the work.

1. Glucocorticoid is the key molecule in this work. The molecular structure should be provided. We have added the molecular structure of dexamethasone - as the most common glucocorticoid used - to the Graphical Abstract (attached as a pdf for convenience).

2. Although this work focuses on studying and analyzing the effect of glucocorticoids on COVID-19 patients, the molecular mechanism of how the molecule affects can be mentioned.

We have added an overview of the main mechanisms to the Introduction, and also a reference for a more detailed discussion of glucocorticoid binding to GRα and onward trans-repression, trans-activation and non-genomic mechanisms. We have kept this addition concise given that the actual mechanisms are well described in the literature, for example in the added reference.

3. As the authors already mentioned, the SARS-CoV-2 variant may affect the results. Their data were collected at the time when new variants (such as alpha and delta) showed in the UK. Thus, I suggest this issue and related references(1-3) should be mentioned in the introduction, not (only) at the end. 

We have added the point on variants to the Introduction together with the related references that were helpfully suggested by the reviewer.

4. For Figure 4, the items can be arranged in a better order and shape.

This figure was produced in Cytoscape. We have slightly modified the figure to try to meet the reviewer's concerns, but have erred on the side of keeping the figure as readable as possible.

Reviewer 4 Report

The work entitled { Multi-omics reveals mechanisms of partial modulation of COVID-19 dysregulation by glucocorticoid treatment} is well written however it still need some minor modifications before acceptance for publication.

At beginning, please note that multiomics is referred to genomics, proteomics, ametabolomics, and transcriptomics, so it is collective techniques not only two.

The authors need to mentions more about the similar previous works in introduction.

in the results section, please mention all proteins lists identified 

Author Response

The work entitled {Multi-omics reveals mechanisms of partial modulation of COVID-19 dysregulation by glucocorticoid treatment} is well written however it still needs some minor modifications before acceptance for publication.

We thank the reviewer for the overall positive response, and also for the small, but nonetheless helpful, improvements. We have amended the manuscript to accommodate the requested changes, as detailed in full below.

1. At beginning, please note that multiomics is referred to genomics, proteomics, ametabolomics, and transcriptomics, so it is collective techniques not only two.

We have amended the introduction to make clear that in this study metabolomics and proteomics are employed. We have also added text to the limitations paragraph to make it explicitly clear that "It should be noted that multi-omics can include the full range of genomics, transcriptomics, proteomics and metabolomics, but in this work proteomics and metabolomics were employed". We hope that this meets the reviewer's concerns.

2. The authors need to mention more about the similar previous works in introduction.

We have added a number of new references, in part at other reviewers' request, including on new variants and proteomics studies and on glucocorticoid action in other upper airway diseases.

3. In the results section, please mention all proteins lists identified 

We have included the full list of proteins and metabolites identified in Supplementary Material (Tables S3 and S4), and have clarified the main body text to make clear that the full list is available in this location.